# A Study of Gene Expression Changes in Human Spinal and Oculomotor Neurons; Identifying Potential Links to Sporadic ALS

**DOI:** 10.3390/genes11040448

**Published:** 2020-04-20

**Authors:** Aayan N. Patel, Dennis Mathew

**Affiliations:** 1Davidson Academy, Reno, NV 89503, USA; aayanp@davidsonacademy.unr.edu; 2Department of Biology, University of Nevada, Reno, NV 89557, USA

**Keywords:** ALS, neurodegeneration, GEO2R, excitotoxicity, GABA signaling, *GAD2*, *CALB1*, *GABRE*

## Abstract

Amyotrophic Lateral Sclerosis (ALS) is a neurodegenerative disease that causes compromised function of motor neurons and neuronal death. However, oculomotor neurons are largely spared from disease symptoms. The underlying causes for sporadic ALS as well as for the resistance of oculomotor neurons to disease symptoms remain poorly understood. In this bioinformatic-analysis, we compared the gene expression profiles of spinal and oculomotor tissue samples from control individuals and sporadic ALS patients. We show that the genes *GAD2* and *GABRE* (involved in GABA signaling), and *CALB1* (involved in intracellular Ca^2+^ ion buffering) are downregulated in the spinal tissues of ALS patients, but their endogenous levels are higher in oculomotor tissues relative to the spinal tissues. Our results suggest that the downregulation of these genes and processes in spinal tissues are related to sporadic ALS disease progression and their upregulation in oculomotor neurons confer upon them resistance to ALS symptoms. These results build upon prevailing models of excitotoxicity that are relevant to sporadic ALS disease progression and point out unique opportunities for better understanding the progression of neurodegenerative properties associated with sporadic ALS.

## 1. Introduction

Amyotrophic Lateral Sclerosis (ALS) is a neurodegenerative disease that causes death and the compromised function of motor neurons. The progression of the disease has several complications, including increased muscle weakness and loss of coordination [1]. There are two types of ALS: sporadic (95% of diagnosed cases) and familial (5% of diagnosed cases). Mutations in several genes, including *C9ORF72* (Chromosome 9 open reading frame 72; helps to regulate RNA transcription and translation), *SOD1* (Super oxide dismutase 1; encodes superoxide dismutase, which helps to eliminate harmful reactive oxygen species), and *TDP-43* (TAR DNA-binding protein 43; encodes a transcription factor) have been implicated in both forms of ALS [2,3,4]. While there is considerable overlap among the genes that are associated with both forms of ALS, the underlying causes for sporadic ALS are less clear [5]. Thus, critical evaluations of human gene expression datasets continue to remain a valuable approach for identifying the underlying causes of this disease. 

One hypothesis for the cause of sporadic ALS pertains to excitotoxicity and neuronal death correlated with high synaptic glutamate levels [6,7,8,9]. In support of this hypothesis, roughly 40% of ALS patients were found to have high glutamate levels in their cerebrospinal fluid [10]. Additionally, glutamate excitotoxicity results in high intracellular Ca^2+^ ion concentrations, which, in turn, interfere with homeostatic cellular processes and might lead to neurodegenerative properties that are associated with ALS [11]. However, the mechanisms underlying glutamate excitotoxicity and increases in intracellular Ca^2+^ signaling in the context of this disease remain unclear [12]. Another curious occurrence that was observed among ALS patients is that, while most motor neurons are susceptible to the disease, certain classes of motor neurons, especially those innervating extraocular, pelvic sphincter, and slow limb muscles, exhibit selective resistance [13]. Oculomotor neurons, for instance, remain phenotypically resistant to neurodegeneration and they are almost completely preserved in ALS autopsies [14]. Recent studies have shown that several genes that are differentially expressed in oculomotor neurons as compared to spinal neurons confer upon them neuroprotective properties [15,16]. It has been suggested that oculomotor-specific expression can be utilized to identify candidates that protect vulnerable motor neurons from neurodegeneration [16]. One or more genes that are differentially expressed in oculomotor neurons may confer added protection against glutamate excitotoxicity and elevated intracellular Ca^2+^ signaling.

Poor knowledge of these mechanisms in the context of neurodegeneration prevents a clearer understanding of the causes underlying sporadic ALS. If we better understood these mechanisms, we could learn how glutamate excitotoxicity leads to the progression of neurodegenerative properties that are associated with ALS. We cannot ignore this link to ALS, given that a large percentage of ALS patients have high levels of glutamate in their cerebrospinal fluid. Finally, a better understanding of these mechanisms might aid in designing specific treatments for the disease, which is currently estimated to impact 16,000 people in the US alone with roughly 5000 new cases diagnosed each year (ALS Association; www.alsa.org).

Here, we take advantage of recent developments in the availability of biological datasets as well as NCBI GEO2R methodology, which has become a popular method for comparing the gene expression levels among different biological groups. Similar approaches have been successfully applied in order to analyze differential gene expression profiles of Parkinson’s and Cancer patients [17,18]. We first compare the gene expression profiles of spinal tissue samples from control individuals with those from sporadic ALS patients. Next, we compare the gene expression profiles of spinal tissue samples and oculomotor samples from neurologically normal patients. Our rationale was that genes that are common and differentially expressed across both experimental comparisons may be relevant in conferring resistance to excitotoxicity in oculomotor neurons in sporadic ALS patients, as well as playing a neuroprotective role in healthy individuals.

## 2. Materials and Methods

### 2.1. Data Collection

The two datasets that were discussed in this study were previously deposited in NCBI’s Gene Expression Omnibus [19] and are accessible through GEO (https://www.ncbi.nlm.nih.gov/geo/) series accession numbers: GSE833 and GSE40438. The first dataset, GSE833, was generated from a study that compared post-mortem samples of control, healthy spinal tissue with samples from both familial and sporadic ALS-affected individuals. The second dataset, GSE40438, was used to compare post-mortem samples of oculomotor tissue with samples of lumbospinal tissue from four healthy subjects. For both datasets, gene expression was measured through microarray analysis. The GSE833 dataset analyzed the total RNA extracted from postmortem gray matter of lumbar spinal cord from 11 individuals, including five with sporadic-ALS (two samples were confirmed to have mutations associated with familial ALS and were excluded from this study) and four normal controls. Microarray analysis, including quantification and normalization of gene expression levels, was conducted using Affymetrix GeneChip 3.1 software [20]. The GSE40438 dataset analyzed the total RNA extracted from brain and spinal cord tissue from neurologically normal human control subjects. Four spinal tissue samples from the subjects designated as the control and four corresponding oculomotor samples were assigned to the experimental. The Bioconductor package PUMA, a Bayesian probabilistic model for probe-level analysis, was used to carry out the normalization and differential expression analysis [21].

### 2.2. GEO2R Analysis and Volcano Plot Construction

We used GEO2R (http://www.ncbi.nlm.nih.gov/geo/geo2r/), an online analysis tool that compares gene expression from groups in a GEO dataset, to identify the differentially expressed genes in GSE833 and GSE40438. The selected metrics for data interpretation included *p*-value, logFC, gene ID, gene name, gene function, and GO process. Data that were obtained from GEO2R for each dataset were converted to text file format and then imported into R (http://www.R-project.org/) to be stored as a data matrix. Volcano plots were created for each dataset at an α-level of 0.05 and a logFC cutoff of 1.3 using the EnhancedVolcano package (https://github.com/kevinblighe/EnhancedVolcano). Using sorting functions in R, genes that met both the α- and logFC cutoffs for each dataset were exported to excel. Common genes across both datasets were identified using the list comparison tool. Only genes that met both cutoffs and were common and differentially expressed across both datasets were used in functional enrichment analysis. For example: genes were upregulated in oculomotor neurons but downregulated in sporadic ALS patients relative to the spinal control and vice versa. 

### 2.3. Functional Enrichment Analysis with STRING and KEGG

These common, differentially expressed genes were then inputted into the STRING Protein Interactome (https://string-db.org/) to gain a stronger understanding of certain enriched functions that they belonged to. The STRING database compiles direct and indirect interactions among proteins from various sources and generates a diagram showing significant protein–protein interactions (PPIs) (Appendix A). Gene ontology resources, such as GeneCards (https://www.genecards.org/) and the Human Protein Atlas (https://www.proteinatlas.org/), were used to find parallels between the functions of these genes and potential cellular pathways that are compromised in ALS and that are important in oculomotor neuron function. Once the collections of genes with apparent enriched PPIs were identified, their respective pathways were further analyzed using the Kyto Encyclopedia of Genes and Genomes (KEGG, https://www.genome.jp/kegg/) database, which helped to find parallels between enriched oculomotor and ALS pathways.

## 3. Results

### 3.1. Differential Gene Expression in Sporadic ALS-Affected Spinal Tissue

First, we examined the genes that are differentially expressed in the spinal tissues of ALS patients as compared to those of healthy individuals. Spinal neurons are a strong control group because they represent a primary set of CNS neurons that are affected in ALS. We performed GEO2R analysis of GSE833, which was obtained in a study that compared the post-mortem samples of control, healthy spinal tissue with samples from both familial and sporadic ALS-affected individuals [20]. We analyzed 7070 genes. Figure 1 presents these data as a volcano plot. This initial analysis returned 357 genes that were differentially expressed (200 genes overexpressed and 157 genes under expressed) between the spinal tissue samples of sporadic ALS patients and healthy individuals. Appendix A presents the top 25 over-expressed genes and the top 25 under-expressed genes in this analysis.

### 3.2. Differential Gene Expression in Oculomotor Neurons of Healthy Patients

We hypothesized that differential gene expression in oculomotor neurons as compared to spinal neurons confers neuroprotective properties to oculomotor neurons in sporadic ALS individuals. We examined the genes that are differentially expressed in the post-mortem spinal tissues and oculomotor tissue samples of healthy individuals to test this hypothesis. We performed another GEO2R analysis using the GSE40438 data set. This data set was used to compare the gene expression of healthy oculomotor neurons and lumbospinal neurons from post-mortem tissue. We analyzed 12,960 genes. Figure 2 presents these data as a volcano plot. This initial analysis returned 4629 genes that were differentially expressed (1340 genes overexpressed and 3289 genes under expressed) between the spinal tissue samples and oculomotor tissue samples of neurologically normal individuals. Appendix A presents the top 25 over-expressed genes and the top 25 under-expressed genes in this analysis.

### 3.3. Identifying Common Genes across Both Datasets

We wondered whether there is a common set of genes whose expression is downregulated in the diseased neurons but upregulated in oculomotor neurons relative to healthy spinal neurons since oculomotor neurons in ALS patients are resistant to neurodegeneration. In order to answer this question, we sought to identify common genes that were differentially expressed in the two datasets: GSE833 and GSE40438. We predicted that identifying such genes may also provide clues as to how gene expression changes in oculomotor neurons may confer them with neuroprotective properties. 

Genes that were screened to meet both the *p*-value cutoff of 0.05 and logFC cutoff of 1.3 in each dataset were stored as vectors in R. We found 70 differentially expressed genes to be common across both datasets with the given *p*-value and logFC cutoffs. For each of these genes, the logFC values from each study were further analyzed. Of these 70 genes, 39 were found to be overexpressed in sporadic ALS neurons and under expressed in oculomotor neurons or vice versa, relative to the expression in control spinal neurons. The other 31 genes were either both upregulated or downregulated in oculomotor and sporadic ALS tissue. Only the 39 differentially regulated genes were analyzed to find explanatory mechanisms for oculomotor evasion of sporadic ALS. 

Subsequently, we analyzed these 39 differentially expressed genes using enriched protein–protein interaction (PPI) and gene ontology analysis. These genes were inputted into theSTRING-interactome to find plausible PPIs (Appendix A). The genes were also individually studied through GeneCards to better support conclusive findings and find enriched functions among them. We also used the KEGG pathway database to find parallels between enriched functions and pathways that are associated with sporadic ALS progression. We especially focused on genes that are implicated in glutamate excitotoxicity as well intracellular Ca^2+^ homeostasis mechanisms due to their relevance for sporadic ALS progression. We noted enriched PPI between *GAD2* (gene encoding Glutamate Decarboxylase 2; an enzyme that is involved in catalyzing the conversion of glutamate to GABA) and *GABRE* (gene encoding GABA Amino Butyric Acid Type A Receptor Subunit Epsilon; the ε-subunit of the GABA_A_ receptor that initiates an inhibitory postsynaptic potential in the CNS). Furthermore, through gene ontology analysis, we found that *CALB1* (gene encoding Calbindin 1; a protein that assists with intracellular calcium ion buffering) is an important gene that is involved in intracellular Ca^2+^ homeostasis. 

All three genes were under expressed in the spinal tissue from sporadic ALS patients compared to spinal tissue from healthy individuals (Figure 3, *GAD2* logFC = −1.78, *GABRE* logFC = −1.39, *CALB1* logFC = −1.96) and overexpressed in oculomotor tissue when compared to spinal tissue in neurologically normal patients (Figure 4, *GAD2* logFC = 2.69, *GABRE* logFC = 2.34, *CALB1* logFC = 2.45). These results suggest that the increased expression levels of GAD2, GABRE, and CALB1 may enable oculomotor neurons to be better equipped to handle potential glutamate excitotoxicity as well as maintain intracellular Ca^2+^ levels, which could explain their reduced susceptibility to neurodegeneration. 

## 4. Discussion

We conducted bioinformatic-analyses of publicly available gene expression datasets of sporadic ALS patients and show that the genes *GAD2*, *GABRE*, and *CALB1* are downregulated in the spinal tissues of sporadic ALS patients when compared to those of the control individuals. However, these three genes are naturally upregulated in oculomotor tissues when compared to the spinal tissues of neurologically normal individuals. *GAD2* and *GABRE* encode proteins that are integral to GABA signaling in the CNS, while *CALB1* encodes a protein that regulates intracellular Ca^2+^ ion levels. Our results strongly suggest that the downregulation of these genes and processes in spinal tissues are related to sporadic ALS disease progression and that the natural upregulation of these genes and corresponding functions in oculomotor neurons confer upon them resistance to ALS symptoms.

GAD2 is a glutamate decarboxylase that catalyzes the conversion of glutamate to GABA. High levels of glutamate, an excitatory neurotransmitter, results in the overstimulation of CNS neurons, leading to excitotoxicity, inflammation, and cell death. It is essential to maintain low levels of glutamate by converting it into GABA in order to prevent excitotoxicity [22]. The importance of glutamate decarboxylase to neuronal excitotoxicity was supported in a recent study where reduced levels of the enzyme in flies mutant for a transcription factor, TDP-43, led to poor locomotion and high excitotoxicity in the flies. Mutations in *TDP-43* are believed to be involved in ALS [23]. The *TDP-43* mutant phenotypes were alleviated by genetically rescuing *GAD2* expression [24]. The higher glutamate levels observed in ALS patients have been suggested as one possible mechanism underlying neurodegeneration. Our results showing that *GAD2* expression levels are lower in sporadic ALS-spinal tissues, possibly leading to reduced conversion of glutamate to GABA in these tissues, support this theory. 

GABA_A_-receptors are critical for the initiation of an inhibitory postsynaptic potential in CSN neurons [25]. Indeed, Brockington and colleagues have suggested that the oculomotor upregulation of GABA_A_-receptor subunits results in increased GABA transmission, which protects against excessive neuronal excitation [21]. Our study specifically finds that *GABRE*, a gene that encodes the ε–subunit of GABA_A_-receptor, is downregulated in sporadic ALS-spinal tissues. Neeland and colleagues suggested that ε-containing GABA_A_ receptors permit higher spontaneous channel gating [26]. Indeed, the level of spontaneous activity of GABA_A_ receptors is causally related to sensitivity and ligand efficacy [27], which further suggests a role for GABRE in the positive modulation of GABA transmission. Thus, the reduction in GABRE expression levels might further enhance the excitotoxic effects of elevated glutamate levels. 

CALB1 belongs to the calbindin class of proteins and plays a role in buffering the intracellular Ca^2+^ levels [28]. Previous studies have suggested that high levels of excitotoxicity in ALS patients may elevate intracellular Ca^2+^ levels in postsynaptic neurons, which in turn could lead to cell death [29]. Patai and colleagues describe that glutamate-induced elevation of intracellular Ca^2+^ levels activate cytoplasmic Ca^2+^-dependent apoptotic proteins as well as mitochondrial dysfunction [30]. Our results showing that the *CALB1* expression levels are lower in sporadic ALS-spinal tissues would suggest that sporadic ALS-affected neurons are less able to buffer increases in intracellular Ca^2+^ levels and, thus, lend support to the theory. In support of our findings, Alexianu and colleagues found that motor neuron populations lost early during ALS progression had little to no expression of calbindin-D_28k_, an alias to CALB1 [31]. 

On the other hand, oculomotor neurons are specialized to endure rapid firing, resulting in enhanced mechanisms against excitotoxicity. Torres-Torrelo and colleagues suggest that oculomotor neurons depend on phasic firing to execute saccadic eye movements. They found that glutamate is crucial in modulating such phasic firing rates [32]. Their results suggest that oculomotor neurons are well-equipped to conduct glutamate phasic firing. The naturally high levels of GAD2 and GABRE expression in these neurons may be required for quickly catalyzing the transformation of Glutamate to GABA and then inducing GABA signaling to elicit an inhibitory post-synaptic potential. Therefore, the oculomotor upregulation of genes that are involved in phasic firing might be required to counter excitotoxicity due to constant and rapid activity and might provide enhanced neuroprotective properties to oculomotor neurons as compared to other motor neurons. In addition, higher expression of CALB1 might help to buffer natural increases in the intracellular Ca^2+^ levels due to the rapid activity of these neurons. Other reasons may also contribute towards oculomotor resistance during ALS. Nizzardo and colleagues show that higher expression of SYT13 (Synaptotagmin 13) in oculomotor neurons is neuroprotective [15]. Allodi and colleagues show that higher levels of IGF-2 (insulin-like growth factor 2) and IGF-1 receptors in oculomotor neurons during ALS could play a role in oculomotor resistance. They further go on to suggest that oculomotor-specific expression can be utilized in order to identify the candidates that protect vulnerable motor neurons from degeneration [16].

We acknowledge that ALS is a multifactorial disease that involves several pathways that have been hypothesized to contribute towards neurodegeneration symptoms. While mutations in *SOD1*, *C97ORF72*, and *TDP-43* have been implicated in familial ALS, the underlying causes for sporadic ALS remain unclear. Comparing the gene expression profiles in diseased tissue samples with control samples is an effective method for identifying molecular players that are associated with disease progression or protection against disease symptoms. Our findings that the downregulation of *GAD2*, *CALB1*, and *GABRE* genes in the spinal tissues of sporadic ALS patients suggest that glutamate excitotoxicity and inefficient buffering of intracellular Ca^2+^ ion levels may lead to the neurodegeneration of spinal tissues in sporadic ALS patients. Our results also suggest that elevated expression levels of the same genes in oculomotor tissues may lead to enhanced mechanisms that allow for oculomotor neurons to evade ALS-related neurodegeneration. 

Our experimental design involved the analysis of GSE40438 and GSE833 to determine genes of interest. There are no expression profiles in the GEO database of oculomotor neurons from ALS patients. Because of this, the observed oculomotor upregulation of genes relative to spinal neurons in healthy patients might not have been significant in patients who were affected by sporadic ALS. Furthermore, the datasets we analyzed in this study are from post-mortem samples, implying that the observed gene expression might not be causally related to disease progression. Additionally, small groups of samples were analyzed in the original studies. GEO2R Analysis is highly dependent upon the availability of biological datasets within the GEO database. Based on the experimental design of our study, GSE833 and GSE40438 were the only datasets that compared the groups of interest, were from human samples, and contained post-mortem spinal samples as a common pivot, control group. We included both a *p*-value and logFC cutoff to screen for significantly differentially expressed genes to reduce the probability of Type 1 error in our study. We also note that other studies have proposed alternate ALS-neurodegeneration models related to excitotoxicity arising from high levels of GABA. Caruchio and colleagues found that C93A cells that are used as a genetic model for ALS display increased levels of GABA_A_-receptor subunit-α1. They suggest that high levels of GABA transmission results in higher Cl^−^ ion influx, leading to osmotic imbalance across the neuronal membranes, ruptures, and lysis [33]. While our study does not necessarily contradict this model, we caution against over interpretation of our results. While our study might point towards important biomarkers for sporadic ALS, we propose that more studies would be needed in order to gain a stronger understanding of specific regulatory mechanisms involving GAD2, CALB1, and GABRE and their relationship to neuroprotective function and disease. 

Overall, our bioinformatic analysis builds upon prevailing models of excitotoxicity that are relevant to sporadic ALS disease progression. It highlights and offers unique opportunities to better understand how high glutamate excitotoxicity and poor buffering of intracellular Ca^2+^ levels lead to the progression of neurodegenerative properties that are associated with sporadic ALS. A better understanding of these mechanisms offers the promise of potential treatments for this disease.

## Figures and Tables

**Figure 1 genes-11-00448-f001:**
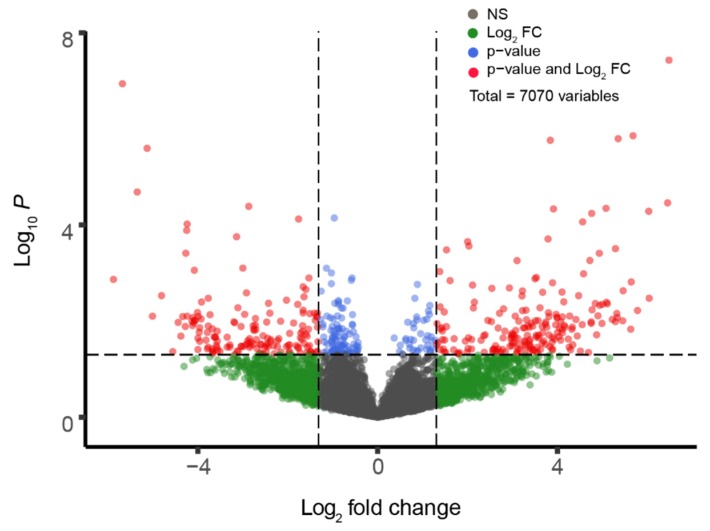
Gene expression comparison between healthy- and Amyotrophic Lateral Sclerosis (ALS)-spinal tissue. Volcano Plot depicting differentially expressed genes between Healthy spinal tissue and ALS-affected grey matter tissue from GSE833, discriminated based on *p*-value and log2(fold-change) at an α level of 0.05 and logFC cutoff of 1.3. Colored dots correspond to individual genes whose expression differences were significant based on both p and logFC value (red dots), only *p*-value (blue dots), only logFC (green dots), or not significant (grey dots) in either.

**Figure 2 genes-11-00448-f002:**
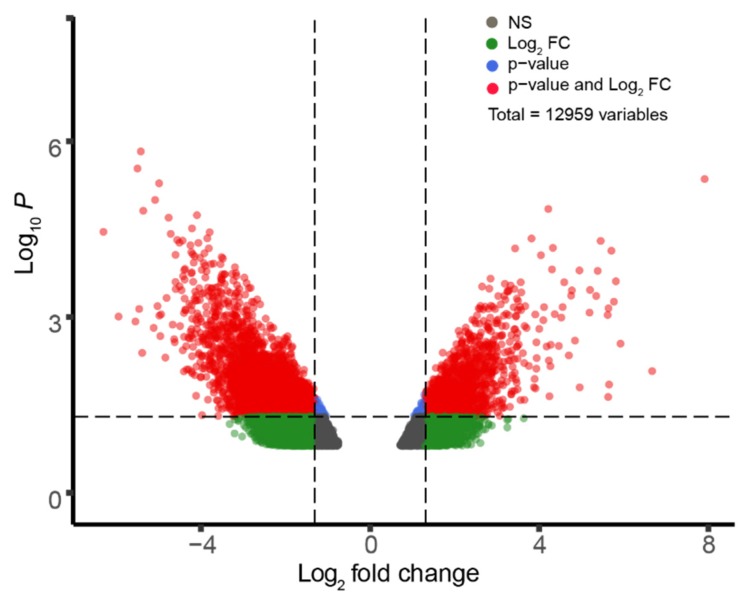
Gene expression comparison between oculomotor and spinal tissue from healthy subjects. Volcano Plot depicting differentially expressed genes between Oculomotor and Lumbospinal tissue from GSE40438, discriminated based on *p*-value and log2 (fold-change) at an α level of 0.05 and logFC cutoff of 1.3. The colored dots correspond to individual genes whose expression differences were significant based on both *p* and logFC value (red dots), only *p*-value (blue dots), only logFC (green dots), or not significant (grey dots) in either.

**Figure 3 genes-11-00448-f003:**
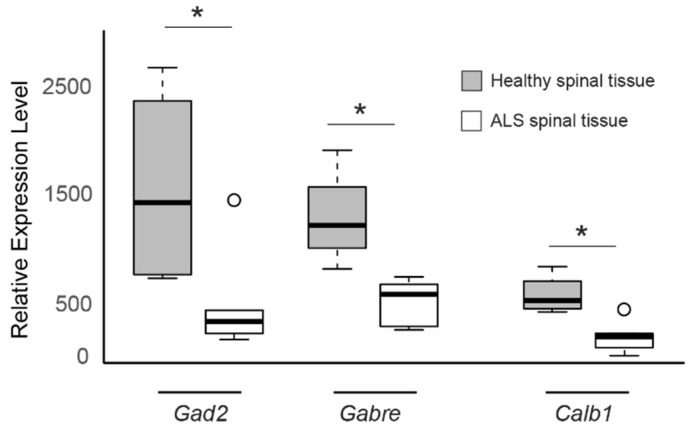
The expression levels of *GAD2*, *GABRE*, and *CALB1* genes are low in sporadic ALS-spinal tissue. Boxplot comparisons of relative gene expression level of *GAD2*, *GABRE*, and *CALB1* in GSE833, comparing expression in control spinal neurons and ALS-affected spinal neurons. Boxes are interquartile ranges. Bars are the non-outlier range as defined by R software (version 3.6.1, R Foundation for Statistical Computing, Vienna, Austria). * *p* < 0.05.

**Figure 4 genes-11-00448-f004:**
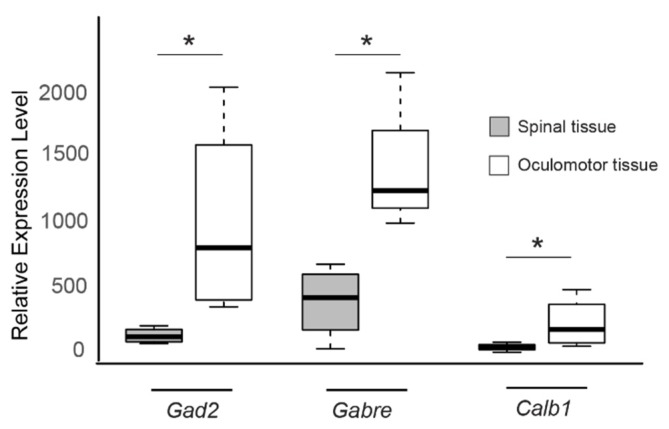
Expression levels of *GAD2*, *GABRE*, and *CALB1* genes are high in oculomotor tissue. Boxplot comparisons of relative gene expression level of *GAD2*, *GABRE*, and *CALB1* in GSE40438, when comparing expression in control lumbospinal neurons and oculomotor neurons. Boxes are interquartile ranges. Bars are the non-outlier range as defined by R software. * *p* < 0.05.

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
