# Peer review of "A Study of Gene Expression Changes in Human Spinal and Oculomotor Neurons; Identifying Potential Links to Sporadic ALS"

_genes, 2020, doi:10.3390/genes11040448_

Round 1
Reviewer 1 Report
The authors compared existing gene expression data sets from spinal tissue of sporadic ALS patients vs. healthy donors and oculomotor tissue vs. spinal tissue from healthy donors.
No information is provided about the microarray analyzes originally used. The number of samples included in the study appears very small. However, the research design is clearly structured and perspicuously. With the exception of the discussion, the manuscript is very readable.
The discussion needs a proofreading, for example line 257 and 258 “While our studies..” or line 226, 228, 243, 245, 255 “suggest/ed that..”
Regarding the small groups of samples analyzed, please include perspectives (marker validation?) in the discussion.
Which are the differentially expressed genes across both datasets? It would be interesting which genes were inputted to further analysis. Please include more supporting information.
Author Response
RESPONSES TO REVIEWER COMMENTS
We are grateful to all three reviewers for the careful critique of our manuscript. Overall, the reviewers were positive about our submission and found our work to be interesting and well-executed. Reviewer 1 mentioned that “the research design is clearly structured and perspicuously. With the exception of the discussion, the manuscript is very readable” and Reviewer 2 mentioned that “The article is well written, and the bioinformatics analysis approach is well thought of”. Reviewer 3 mentioned that “Overall, the manuscript is well written and organized. … The evidence that oculomotor neurons, unlike spinals, remains functionally active through disease progression are interesting and could be really useful to identify new target genes and further therapeutic investigation”.
The reviewers had many excellent suggestions to further improve the manuscript. The co-authors have given serious consideration to each comment, and we have incorporated almost all the suggestions into our revised manuscript. Detailed responses to each comment are presented below in blue. Corresponding changes in the manuscript are also highlighted in blue.
Responses to Reviewer 1’s Comments and Suggestions for Authors
The authors compared existing gene expression data sets from spinal tissue of sporadic ALS patients vs. healthy donors and oculomotor tissue vs. spinal tissue from healthy donors.
1) No information is provided about the microarray analyzes originally used. The number of samples included in the study appears very small. However, the research design is clearly structured and perspicuously. With the exception of the discussion, the manuscript is very readable.
We have now provided more information and appropriate references for the original microarray analyses that were used. See Lines 82-92.
2) The discussion needs a proofreading, for example line 257 and 258 “While our studies..” or line 226, 228, 243, 245, 255 “suggest/ed that..”
We have made the edits indicated by the reviewer. Additionally, we have proofread the entire manuscript as per the reviewer’s suggestion.
3) Regarding the small groups of samples analyzed, please include perspectives (marker validation?) in the discussion.
We thank the reviewer for this suggestion. As per the reviewer’s suggestion, we have provided in the discussion, our perspective and approach to reduce the probability of Type I errors in our study. See lines 253-258
4) Which are the differentially expressed genes across both datasets? It would be interesting which genes were inputted to further analysis. Please include more supporting information.
We have now provided, as Supplementary figures, a list of the top 25 over-expressed genes and the top 25 under-expressed genes for each of the analysis shown in Figures 1 and 2. Please see Supplementary Table 1 and Supplementary Table 2. The over-expressed genes are in blue font and the under-expressed genes are in red font. We have also provided a STRING-interactome network showing protein-protein interactions related to the 39 (common differentially expressed) genes of interest across both datasets in our study including GAD2, GABRE, and CALB1. See Supplementary figure 1.
Reviewer 2 Report
In this article, the authors investigate the difference in gene expression in spinal and oculomotor tissue of healthy individuals and ALS patients. Since oculomotor neurons are spared from toxicity in ALS patients, the authors hypothesize that certain genetic alterations in this tissue type might have a protective effect in ALS. Through bioinformatic analysis, the authors find that genes GAD2, GABRE and CALB1 are downregulated in spinal tissue and overexpressed in oculomotor tissue of ALS patients. Hence the authors conclude that these three genes migt be involved in conferring neuroprotection to oculomotor neurons in ALS patients.
The article is well written and the bioinformatics analysis approach is well thought of. However, my major comment is that the authors must experimentally show that the three genes they identified are in fact important in neuroprotection in ALS.
Under minor comments, the authors must include atleast the top 50-100 over and underexpressed genes found in Figure 1 and 2.
Author Response
RESPONSES TO REVIEWER COMMENTS
We are grateful to all three reviewers for the careful critique of our manuscript. Overall, the reviewers were positive about our submission and found our work to be interesting and well-executed. Reviewer 1 mentioned that “the research design is clearly structured and perspicuously. With the exception of the discussion, the manuscript is very readable” and Reviewer 2 mentioned that “The article is well written, and the bioinformatics analysis approach is well thought of”. Reviewer 3 mentioned that “Overall, the manuscript is well written and organized. … The evidence that oculomotor neurons, unlike spinals, remains functionally active through disease progression are interesting and could be really useful to identify new target genes and further therapeutic investigation”.
The reviewers had many excellent suggestions to further improve the manuscript. The co-authors have given serious consideration to each comment, and we have incorporated almost all the suggestions into our revised manuscript. Detailed responses to each comment are presented below in blue. Corresponding changes in the manuscript are also highlighted in blue.
Responses to Reviewer 2’s Comments and Suggestions for Authors
In this article, the authors investigate the difference in gene expression in spinal and oculomotor tissue of healthy individuals and ALS patients. Since oculomotor neurons are spared from toxicity in ALS patients, the authors hypothesize that certain genetic alterations in this tissue type might have a protective effect in ALS. Through bioinformatic analysis, the authors find that genes GAD2, GABRE and CALB1 are downregulated in spinal tissue and overexpressed in oculomotor tissue of ALS patients. Hence the authors conclude that these three genes migt be involved in conferring neuroprotection to oculomotor neurons in ALS patients.
1) The article is well written, and the bioinformatics analysis approach is well thought of. However, my major comment is that the authors must experimentally show that the three genes they identified are in fact important in neuroprotection in ALS.
I hope the reviewer will understand that undertaking experiments to show that the three genes identified in this study are in fact important for neuroprotection in ALS is beyond the scope of this bioinformatic analysis and our lab’s capability. To do so, we would have to initiate new collaborations with appropriate researchers and optimize techniques, which would be costly in terms of both time and resources.
However, to address the reviewer’s concerns in a different way, we have provided additional references and information about the validity of our approach. See lines 50-54, lines 231-237. We have also emphasized in the Discussion section that further experimental validation would be required to conclusively implicated GAD2, GABRE, and CALB1 in neuroprotective functions during ALS. See lines 264-267.
2) Under minor comments, the authors must include at least the top 50-100 over and underexpressed genes found in Figure 1 and 2.
We have now provided, as Supplementary figures, a list of the top 25 over-expressed genes and the top 25 under-expressed genes for each of the analysis shown in Figures 1 and 2. Please see Supplementary Table 1 and Supplementary Table 2. The over-expressed genes are in blue font and the under-expressed genes are in red font.
Reviewer 3 Report
In this manuscript, Patel and Mathew used GEO2R methodology and biological free datasets to analyze differential gene expression profiles of spinal and oculomotor post-mortem tissues from sporadic ALS patients and healthy neurological controls. Since in ALS, spinal motor neurons degenerate, but some motor neuron subtypes are spared, including oculomotor neurons, the rational of this study was that genes that are common and differentially expressed may be relevant in conferring resistance in oculomotor neurons in ALS patients as well as playing a neuroprotective role in healthy subjects.
In particular, authors showed, only based on bioinformatic analysis, that three genes (GAD2, GABRE, and CALB1), involved in glutamate excitotoxicity and Ca2+ homeostasis, were significantly downregulated in spinal tissues of ALS patients compared to controls and upregulated in oculomotor tissue versus spinal tissue of healthy subjects. Their results suggest that downregulation of these genes and processes in spinal tissues are related to ALS disease progression and that their upregulation in oculomotor neurons can confer resistance to ALS symptoms.
Overall, the manuscript is well written and organized. The English language and style are adequate. The evidence that oculomotor neurons, unlike spinals, remains functionally active through disease progression are interesting and could be really useful to identify new target genes and further therapeutic investigation. However, only bioinformatic analysis were not sufficiently to support the hypothesis, that is only a supposition without any effective data. Nevertheless, the lacking of data on ALS oculomotor tissues weaken the conclusion.
Here are the major points to be considered by the authors:
- Validation of genes (GAD2, GABRE, AND CALB1) expression in patient post mortem tissue or in an in vitro sporadic ALS model (iPSC-derived lines) must be added to support bioinformatic data (RT-PCR, western blot, or ICC/IHC).
- It can be interesting to add further bioinformatic analysis on pathways related to identified genes. Moreover, STRING-interactome network inserted as graph can be useful.
- In Materials and Methods section, “Data collection paragraph” is not clear and patient data should be defined better.
- Introduction/discussion: The authors reported some important studies (Hedlund and Shaw papers) regarding motor neuron vulnerability and oculomotor resistance in amyotrophic lateral sclerosis. However, the recently paper of Corti’s group regarding the same topic is lacking. This study should be added and discussed.
- Healthy controls data from GSE833 dataset were compared only with sporadic ALS patients (and none familiar), even if this GSE833 contains both familiar and sporadic data. Is it correct? The author should specify this information in all the manuscript. “ALS” should be modified in sporadic ALS.
- Line 78: “In GSE40438, the four spinal tissue samples from the patients were designated as the control while the four corresponding oculomotor samples were assigned to the experimental”. Considering that the dataset GSE40438 was used to compare post-mortem samples of oculomotor tissue with samples of lumbospinal tissue from four neurologically normal patients, the authors should specify normal patient or healthy subject.
Minor point:
- Introduction: clinical references should be always added for these sentences (lines 30/31) and also for the most common mutated genes related to ALS.
- Human gene symbols should contain italicized characters that are all in upper -case. Please recheck all the manuscript, also in figures and figure legends.
- Gene symbol should be explicit for the first time in the manuscript. Please recheck.
- Figure 2. The title of figure legend should be modified into “Gene expression comparison between oculomotor and spinal tissue from healthy subjects”.

Author Response
RESPONSES TO REVIEWER COMMENTS
We are grateful to all three reviewers for the careful critique of our manuscript. Overall, the reviewers were positive about our submission and found our work to be interesting and well-executed. Reviewer 1 mentioned that “the research design is clearly structured and perspicuously. With the exception of the discussion, the manuscript is very readable” and Reviewer 2 mentioned that “The article is well written, and the bioinformatics analysis approach is well thought of”. Reviewer 3 mentioned that “Overall, the manuscript is well written and organized. … The evidence that oculomotor neurons, unlike spinals, remains functionally active through disease progression are interesting and could be really useful to identify new target genes and further therapeutic investigation”.
The reviewers had many excellent suggestions to further improve the manuscript. The co-authors have given serious consideration to each comment, and we have incorporated almost all the suggestions into our revised manuscript. Detailed responses to each comment are presented below in blue. Corresponding changes in the manuscript are also highlighted in blue.
Responses to Reviewer 3’s Comments and Suggestions for Authors
In this manuscript, Patel and Mathew used GEO2R methodology and biological free datasets to analyze differential gene expression profiles of spinal and oculomotor post-mortem tissues from sporadic ALS patients and healthy neurological controls. Since in ALS, spinal motor neurons degenerate, but some motor neuron subtypes are spared, including oculomotor neurons, the rational of this study was that genes that are common and differentially expressed may be relevant in conferring resistance in oculomotor neurons in ALS patients as well as playing a neuroprotective role in healthy subjects.
In particular, authors showed, only based on bioinformatic analysis, that three genes (GAD2, GABRE, and CALB1), involved in glutamate excitotoxicity and Ca2+ homeostasis, were significantly downregulated in spinal tissues of ALS patients compared to controls and upregulated in oculomotor tissue versus spinal tissue of healthy subjects. Their results suggest that downregulation of these genes and processes in spinal tissues are related to ALS disease progression and that their upregulation in oculomotor neurons can confer resistance to ALS symptoms.
Overall, the manuscript is well written and organized. The English language and style are adequate. The evidence that oculomotor neurons, unlike spinals, remains functionally active through disease progression are interesting and could be really useful to identify new target genes and further therapeutic investigation. However, only bioinformatic analysis were not sufficiently to support the hypothesis, that is only a supposition without any effective data. Nevertheless, the lacking of data on ALS oculomotor tissues weaken the conclusion.
Here are the major points to be considered by the authors:
1) Validation of genes (GAD2, GABRE, AND CALB1) expression in patient post-mortem tissue or in an in vitro sporadic ALS model (iPSC-derived lines) must be added to support bioinformatic data (RT-PCR, western blot, or ICC/IHC).
I hope the reviewer will understand that doing experiments to show that the three genes identified in this study are in fact important for neuroprotection in ALS is beyond the scope of this bioinformatic analysis and our lab’s capability. To do so, we would have to initiate new collaborations with appropriate researchers and optimize techniques, which would be costly in terms of both time and resources.
However, to address the reviewer’s concerns in a different way, we have provided additional references and information about the validity of our approach. See lines 50-54, lines 231-237. We have also emphasized in the Discussion section that further experimental validation would be required to conclusively implicated GAD2, GABRE, and CALB1 in neuroprotective functions during ALS. See lines 264-267.
2) It can be interesting to add further bioinformatic analysis on pathways related to identified genes. Moreover, STRING-interactome network inserted as graph can be useful.
We have now provided a STRING-interactome network showing protein-protein interactions related to the 39 genes of interest in our study including GAD2, GABRE, and CALB1. See Supplementary figure 1.
3) In Materials and Methods section, “Data collection paragraph” is not clear and patient data should be defined better.
We have now provided more information and appropriate references for the original microarray analyses in the Data collection section of Materials and methods. See Lines 83-92
4) Introduction/discussion: The authors reported some important studies (Hedlund and Shaw papers) regarding motor neuron vulnerability and oculomotor resistance in amyotrophic lateral sclerosis. However, the recently paper of Corti’s group regarding the same topic is lacking. This study should be added and discussed.
We thank the reviewer for pointing out our oversight. We have now introduced two recent papers from Corti’s group (both in introduction and Discussion). Further in support of our study, Corti’s group had predicted that oculomotor-specific expression can be utilized to identify candidates that protect vulnerable motor neurons from degeneration. See lines 50-54 and lines 231-237.
5) Healthy controls data from GSE833 dataset were compared only with sporadic ALS patients (and none familiar), even if this GSE833 contains both familiar and sporadic data. Is it correct? The author should specify this information in all the manuscript. “ALS” should be modified in sporadic ALS.
Yes, the reviewer is correct in that the healthy control data from GSE833 dataset were compared only with data from sporadic ALS patients. We have specified this information in the manuscript. See lines 83-86. Further, we have changed ‘ALS’ to ‘sporadic ALS’ wherever appropriate to keep the narrative consistent with our findings.
6) Line 78: “In GSE40438, the four spinal tissue samples from the patients were designated as the control while the four corresponding oculomotor samples were assigned to the experimental”. Considering that the dataset GSE40438 was used to compare post-mortem samples of oculomotor tissue with samples of lumbospinal tissue from four neurologically normal patients, the authors should specify normal patient or healthy subject.
We have edited the sentence as per the reviewer’s suggestion.
Minor points:
1) Introduction: clinical references should be always added for these sentences (lines 30/31) and also for the most common mutated genes related to ALS.
As per the reviewer’s suggestion, we have now added clinical references for these sentences. See line 37.
2) Human gene symbols should contain italicized characters that are all in upper -case. Please recheck all the manuscript, also in figures and figure legends.
We have gone through the entire manuscript and ensured that all human genes are italicized and that all characters are in upper-case.
3) Gene symbol should be explicit for the first time in the manuscript. Please recheck.
We have gone through the entire manuscript and ensured that all gene symbols are defined when they are mentioned for the first time in the manuscript (except in the abstract for sake of staying within the word count limit).
4) Figure 2. The title of figure legend should be modified into “Gene expression comparison between oculomotor and spinal tissue from healthy subjects”.
As per the reviewer’s suggestion we have modified the title for figure legend #2.
Round 2
Reviewer 2 Report
I am okay with the explanation provided by the authors as well as the corrections they have made. I have no other concerns with the article.
Reviewer 3 Report
The authors did satisfactorily reply to my comments except for the validation of results.